# Magnitude of standard precautions practices among healthcare workers in health facilities of Low and Middle Income Countries: A systematic review and meta-analysis

**Mengistu Yilma** [1]*, **Girma Taye**[1], **Workeabeba Abebe**[2]

1 School of Public Health, Addis Ababa University, Addis Ababa, Ethiopia, 2 School of Medicine, Addis Ababa University, Addis Ababa, Ethiopia

☯ These authors contributed equally to this work.

* newmany55@gmail.com

## Abstract

**Data Availability Statement:** All relevant data are within the manuscript and its Supporting information files.

### Background

Standard precautions are the minimum standard of infection control to prevent transmission of infectious agents, protect healthcare workers, patients, and visitors regardless of infection status. The consistent implementation of standard precautions is highly effective in reducing transmission of pathogens that cause HAIs. Despite their effectiveness, compliance, resources, patient behavior, and time constraints are some of the challenges that can arise when implementing standard precautions. The main objective of this meta-analysis was to show the pooled prevalence of safe standard precaution practices among healthcare workers in Low and Middle Income Countries (LMICs).

### Methods

A systematic review and meta-analysis was conducted for this study. We systematically searched observational study articles from PubMed Central and Google Scholar. We included articles published any year and involving healthcare workers. We used Preferred Reporting Items for Systematic Reviews and Meta-Analysis (PRISMA). The random effect model was used to estimate the pooled prevalence. The meta-analysis, sensitivity analysis, subgroup analysis, and publication bias (funnel plot, and Egger's tests) were conducted.

### Results

A total of 46 articles were included in this study. The pooled prevalence of standard precautions practices among healthcare workers in LMICs was 53%, with a 95% CI of (47, 59). These studies had a total sample size of 14061 with a minimum sample size of 17 and a maximum sample size of 2086. The majority of the studies (82.6%) were conducted in hospitals only (all kinds), and the remaining 17.4% were conducted in all health facilities, including hospitals.

**Funding:** The author(s) received no specific funding for this work.

**Competing interests:** The authors have declared that no competing interests exist.

## Conclusions

The pooled prevalence of standard precautions practices among healthcare workers in LMICs was suboptimal. The findings of this study can have substantial implication for healthcare practice and policy making by providing robust evidence with synthesized and pooled evidence from multiple studies.

## Trial registration

Registered on PROSPERO with record ID: CRD42023395129, on the 9[th] Feb. 2023.

## Background

Standard precautions are infection prevention and control measures that are designed to reduce the risk of infection transmission in healthcare settings, regardless of the severity of the infections. The components of standard precautions include: Hand hygiene, personal protective equipment (PPE), sharps safety, respiratory hygiene, and injection safety, sterilization of instruments and devices, and environmental cleanliness [1]. Historically, infection control was developed during the late 1950s in the United States to control nosocomial staphylococcal infections. The discipline of infection control developed over the next 50 years, focusing on surveillance for healthcare associated infections (HAIs) and then incorporating the science of epidemiology to elucidate risk factors for HAIs. Currently, the focus of hospital programs has shifted from infection control to infection prevention, with the aim of rapid identification of infection and timely actions for infection reduction [2].

The consistent implementation of standard precautions is highly effective in **reducing the transmission of pathogens** to and from healthcare providers, clients, patients, residents, and visitors **that cause HAIs. Despite their effectiveness, compliance among healthcare workers, limited resources, patient behavior, and time constraints are some of the challenges that can arise when implementing standard precautions** [3–5]. Healthcare workers' adherence to IPC practice is important to offering safe and high-quality health care in the healthcare delivery facilities [6]. In recent years, the occurrence of new and emerging health problems like COVID-19 has changed perspectives about IPC practices in healthcare facilities [7].

The infection prevention and control program reduces HAIs so effectively that a study showed a 32% reduction in HAIs in five years of implementation of the IPC program [8]. According to the report from the Organization for Economic Co-operation and Development (OECD) [9], it is estimated that promoting simple IPC measures such as hand hygiene could reduce the AMR burden by about 40%. Improving IPC can reduce the magnitude of non-resistant HAIs, which cause millions of days of hospital stays and huge financial losses [10].

Healthcare associated infections are among the issues worldwide for the health system. HAIs are a significant cause of morbidity and mortality, as well as increased hospitalization costs [11]. Ebola, MERS, COVID-19, and monkey pox are some of the issues that have imposed an additional burden on the health system in recent years. There are single studies and some systematic reviews and meta-analyses conducted in a few countries in the LMICs, but this study summarized existing evidence from multiple articles in the region. Therefore, the current study helps to identify areas where the evidence is lacking, enhance statistical power for generalizability, and improve evidence-based decision-making in the field of infection prevention and control. Thus, the objective of this study was to synthesize evidence and show the pooled prevalence of safe standard precaution practices among healthcare workers in LMICs.

The review provides an answer to the question, "What is the overall and subgroup pooled prevalence of safe standard precautionary practices among healthcare workers in LMICs?"

## Methods

**We conducted a systematic review and meta-analysis using the four fundamental stages of identification, selection, abstraction, and analysis. We identified databases as the source of information for pertinent articles for our systematic review and meta-analysis** We systematically searched articles from PubMed Central and Google Scholar. The list of references was organized by direct search from databases with key words, using a snowballing system, and hand search.

### Search strategy

The protocol for this systematic review and meta-analysis was registered in the International Prospective Register of Systematic Reviews (PROSPERO), the University of York Centre for Reviews and Dissemination (record ID: CRD42023395129, on the 9th Feb. 2023).

The search strategy for PubMed was prepared using keywords from the study topic and searched by combining with the Boolean operators "OR" and "AND" as follows:

("Infection prevention" OR "infection control" OR "cross infection" OR "universal precautions") AND ("health personnel" OR "health worker*" OR "healthcare worker*" OR "healthcare provider*") AND ("Health Facilities" OR Hospital* OR "health center*" OR "public health facilities") AND (Afghanistan OR Guinea OR Rwanda OR Burkina Faso OR Guinea-Bisseau OR Sierra Leone OR Burundi OR Liberia OR Somalia OR Central African Republic OR Madagascar OR South Sudan OR Chad OR Malawi OR Sudan OR DR Congo OR Mali OR Syrian Arab Republic OR Eritrea OR Mozambique OR Togo OR Ethiopia OR Niger OR Uganda OR Gambia OR North Korea OR Yemen OR Albania OR Algeria OR American Samoa OR Angola OR Argentina OR Armenia OR Azerbaijan OR Bangladesh OR Belarus OR Belize OR Benin OR Bhutan OR Bolivia OR Bosnia and Herzegovina OR Botswana OR Brazil OR Bulgaria OR Cambodia OR Cameroon OR Cape Verde OR China OR Colombia OR Comoros OR Costa Rica OR Cuba OR Djibouti OR Dominica OR Dominican Republic OR Ecuador OR Egypt OR El Salvador OR Equatorial Guinea OR Eswatini OR Fiji OR Gabon OR Georgia OR Ghana OR Grenada OR Guatemala OR Guyana OR Haiti OR Honduras OR India OR Indonesia OR Iran OR Iraq OR Ivory Coast OR Jamaica OR Jordan OR Kazakhstan OR Kenya OR Kiribati OR Kyrgyzstan OR Laos OR Lebanon OR Lesotho OR Libya OR Malaysia OR Maldives OR Marshall Islands OR Mauritania OR Mauritius OR Mexico OR Micronesia OR Moldova OR Mongolia OR Montenegro OR Morocco OR Myanmar OR Namibia OR Nepal OR Nicaragua OR Nigeria OR North Macedonia OR Pakistan OR Palestine OR Panama OR Papua New Guinea OR Paraguay OR Peru OR Philippines OR Republic of the Congo OR Romania OR Russia OR Saint Lucia OR Saint Vincent and the Grenadines OR Samoa OR Sao Tome and Principe OR Senegal OR Serbia OR Solomon Islands OR South Africa OR Sri Lanka OR Suriname OR Tajikistan OR Tanzania OR Thailand OR Timor-Leste OR Tonga OR Tunisia OR Turkey OR Turkmenistan OR Tuvalu OR Ukraine OR Uzbekistan OR Vanuatu OR Vietnam OR Zambia OR Zimbabwe)

Based on this search strategy, we retrieved 8,115 articles on date 03/10/2022.

### Selection and management processes

The articles were selected based on the information required: study design, study year, language, and study participants. We used standardized data extraction form designed for this purpose. In the abstraction process, the two authors (MY and GT) independently reviewed

articles and decided to include or exclude them based on the criteria. We used the Rayyan webpage to manage the screening and selection of articles. After the independent screening was completed, we made blind off option available in Rayyan to see the conflicts between the two independent reviewers. The conflicts were resolved by discussion moderated by the third author (WA), Stata version 16 software was used for analysis. Our protocol follows recommendations set out in the statement of Preferred Reporting Items for Systematic Reviews and Meta-Analysis (PRISMA) [12].

The papers' titles and abstracts were used for the preliminary screening of articles. Then, the full-text articles were further examined to determine their inclusion or exclusion from the systematic review and meta-analysis.

## Inclusion criteria

- Study design: observational studies

- Population: studies involving healthcare workers

- Language: articles published in the English language

- Reported condition: studies that reported the prevalence of one or more SP components among healthcare workers

- Availability of full texts

- Year of publication: all years

- Study area: studies conducted in Low and Middle Income Countries

## Exclusion criteria

- Articles assessing SP practices related to COVID-19, Ebola, and other newly emerged diseases were excluded because of its extraordinary practices.

- Articles used direct observation method of data collection were excluded since the real time practice may not be comparable with reported practice

## The outcome of the study

The proportion of standard precautions practices among healthcare workers in LMIC were pooled and reported as a prevalence in this meta-analysis. The random effect model was used to estimate the pooled prevalence of standard precautions. Subgroup analysis was conducted by using publication years, types of practices, study setting, profession and sampling methods as a moderator variables.

## Operational definition

Poor quality article–the article which was evaluated with the Critical Appraisal tool and has low methodological quality. Articles with inadequate sample size, lack of clear inclusion and exclusion criteria, and has high risk of bias in selection considered poor quality based on the Newcastle-Ottawa Scale (NOS).

Safe practices of standard precautions–the overall compliance of healthcare workers with the IPC standard precautions like proper hand hygiene practice, proper utilization of personal

protective equipment, medical equipment processing, proper healthcare waste management, prevention of SSI, and injection safeties.

## Data extraction

The data from the included articles were exported from Rayyan webpage to a CSV file. The following information was extracted: the first author's name, year of publication, study design, study setting, study population, sampling technique, the IPC components assessed, sample size, and prevalence (proportion) of included studies.

## Quality assessment

The quality of the selected articles was assessed using the Newcastle-Ottawa Scale (NOS), customized for a cross-sectional study [13]. This tool has three sections: (1) selection with 5 stars; (2) comparability between groups with 2 stars; and (3) outcome with 3 stars. The maximum possible score was 10 stars, which represented the highest methodological quality.

## Data analysis and synthesis

The extracted data in csv format were imported into Stata 18 software for analysis. The descriptive characteristics of each study were presented in a table. The basic statistics, like the random effect size for each study with the 95% CI, P-values, and heterogeneity, were reported. The forest and funnel plots were also displayed to visualize the analysis output graphically.

We conducted subgroup analyses by publication year, types of IPC practices, study setting, profession, and sampling technique. The purposes of these subgroup analyses were to identify prevalence differences between the specified subgroups, increase statistical power by detecting small prevalence that may not be seen in the overall analysis, and detect heterogeneity differences between subgroups. Publication bias was assessed using a funnel plot. In the absence of publication bias, the plot resembles a symmetrical inverted funnel. Egger's weighted regression was used in checking the publication bias ($P < 0.05$), considered statistically significant. The Galbraith plot was conducted to assess heterogeneity and detect potential outliers. It is also applied to quantify the impact of these potential outliers on the estimation of the overall effect size. We also conducted a leave-one-out sensitivity analysis to assess the influence of each individual study. This sensitivity analysis is used to investigate the effect of each individual study on the overall meta-analysis summary estimates. We used a random effect model for this meta-analysis.

## Results

### Selection and identification of studies

There were 8,115 search results found on PubMed Central based on the designed search strategy. The other 112 articles were identified from Google Scholar. These articles were different from those found in PubMed Central. All these articles (8227) were uploaded to the Rayyan website for screening. From these articles, the Rayyan system identified 211 as possible duplicates; we checked them and let them be deleted; 8016 articles remain after duplicates were removed. Then we conducted a primary screening and excluded 7587 articles that were not in the topic of interest and remained with 429 articles. From these articles, 270 were excluded after further title review, 78 were excluded after abstract review, and 81 remained for full text review. Finally, after the detailed review, we came up with 46 articles that fulfilled the inclusion criteria (Fig 1).

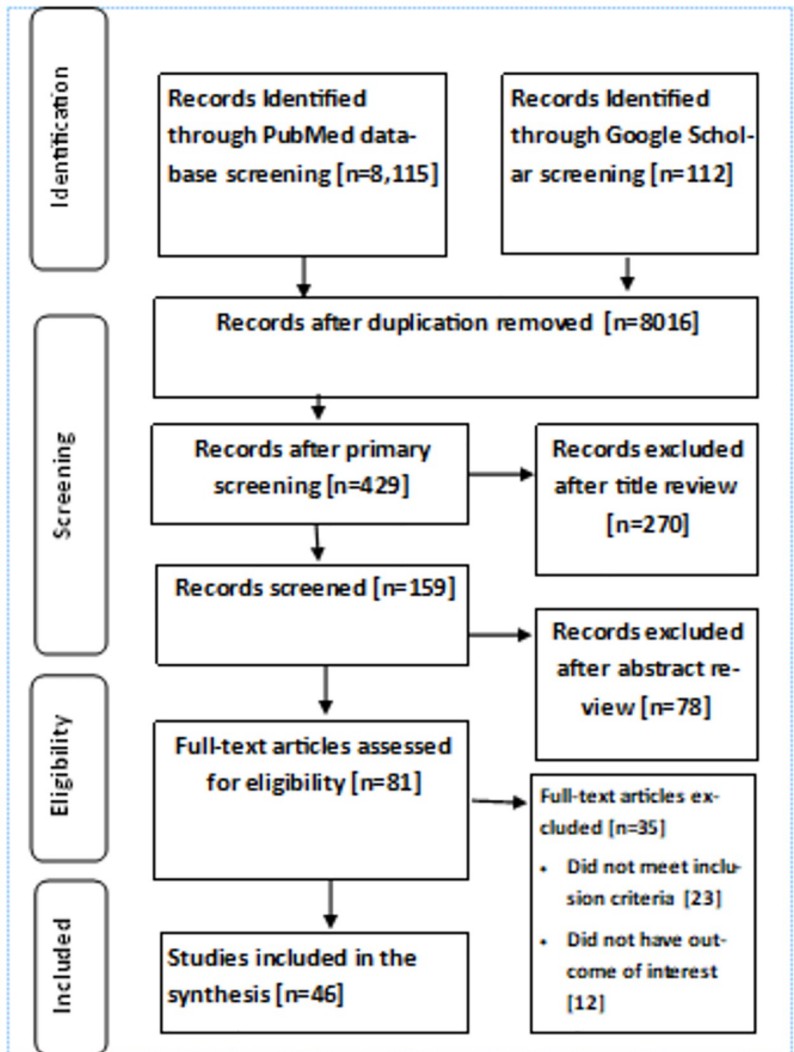

**Fig 1. PRISMA (preferred reporting items for systematic reviews and meta-analyses) flowchart of study inclusions and exclusions.**

## Characteristics of included studies

A total of 46 studies were included for this systematic review and meta-analysis. The overall sample size of these studies was 14061, with a minimum of 17 and a maximum of 2086. The smallest (17 participants) and the largest (2086 participants) were conducted by Kebebe B et al. in Ethiopia and Laraqui O et al. in Morocco, respectively. All of the studies included were observational study design which 96% were cross-sectional. The majority of the studies (82.6%) were conducted in hospitals only, and the remaining 17.4% were conducted in other health facilities including hospitals. The studies were identified from Low-and-Middle-Income Countries (LMIC) where more than half of the studies (58.7%) were identified from Ethiopia. From these standard precautions practice assessments, 41.3% were conducted on hand hygiene only. The participants involved in the studies were all types of professionals (physician, health officer, nurse, midwife, laboratory, and environmental health), with a proportion of 80.4%, while the remaining 19.6% were only nurses. The prevalence of SP practices ranges

**Table 1. Characteristics of included studies for systematic review and meta-analysis on prevalence standard precautions practices among HCWs in Low and Middle Income Countries.**

| Study Name | Year | Country | design | Study setting | Profession | Sampling | Types of IPC practices assessed | Event | sample size | proportion | seproportion | Quality |
|---|---|---|---|---|---|---|---|---|---|---|---|---|
| Ajibola S et al. [14] | 2014 | Nigeria | CS | Hospitals only | All HCWs | Random | All others standard precautions | 23 | 372 | 0.0618 | 0.0124 | 7 |
| Engdaw GT et al. [15] | 2019 | Ethiopia | CS | Hospitals only | All HCWs | Random | Hand hygiene only | 50 | 335 | 0.1492 | 0.0194 | 7 |
| Yohaness T et al. [12] | 2019 | Ethiopia | CS | Hospitals only | All HCWs | Random | All others standard precautions | 41 | 274 | 0.1496 | 0.0215 | 7 |
| Ngwa CH et al. [16] | 2018 | Cameron | CS | Hospitals only | All HCWs | Random | All others standard precautions | 41 | 216 | 0.1898 | 0.0266 | 7 |
| Hosseinialhashemi M et al. [17] | 2015 | Iran | CS | Hospitals only | All HCWs | Non-probability | Hand hygiene only | 121 | 377 | 0.3209 | 0.0240 | 7 |
| Adegboye MB et al. [18] | 2018 | Nigeria | CS | Hospitals only | All HCWs | Non-probability | Hand hygiene only | 26 | 80 | 0.3250 | 0.0523 | 7 |
| Arinze-Onyia SU et al. [19] | 2018 | Nigeria | CS | Hospitals only | All HCWs | Non-probability | Hand hygiene only | 208 | 629 | 0.3306 | 0.0185 | 7 |
| Geberemariyam BS et al. [20] | 2018 | Ethiopia | CS | All HCFs | All HCWs | Non-probability | All others standard precautions | 235 | 648 | 0.3626 | 0.0188 | 7 |
| Tadesse AW et al. [21] | 2020 | Ethiopia | CS | Hospitals only | All HCWs | Random | All others standard precautions | 168 | 422 | 0.3981 | 0.0238 | 7 |
| Gebresillassie A et al. [22] | 2014 | Ethiopia | CS | All HCFs | All HCWs | Non-probability | All others standard precautions | 207 | 483 | 0.4285 | 0.0225 | 7 |
| Umoh VA et al. [23] | 2020 | Nigeria | CS | Hospitals only | All HCWs | Random | All others standard precautions | 24 | 51 | 0.4705 | 0.0698 | 7 |
| Tenna A et al. [24] | 2013 | Ethiopia | CS | Hospitals only | All HCWs | Non-probability | Hand hygiene only | 125 | 261 | 0.4789 | 0.0309 | 7 |
| Hang Pham TT et al. [25] | 2019 | Vietnam | CS | All HCFs | All HCWs | Random | All others standard precautions | 151 | 314 | 0.4808 | 0.0281 | 7 |
| Gulilat K et al. [26] | 2014 | Ethiopia | CS | All HCFs | All HCWs | Random | All others standard precautions | 192 | 354 | 0.5423 | 0.0264 | 8 |
| Assefa J et al. [27] | 2020 | Ethiopia | CS | All HCFs | All HCWs | Random | All others standard precautions | 94 | 171 | 0.5497 | 0.0380 | 7 |
| Desta M et al. [28] | 2018 | Ethiopia | CS | Hospitals only | All HCWs | Non-probability | All others standard precautions | 86 | 150 | 0.5733 | 0.0403 | 7 |
| Yazie TD et al. [29] | 2019 | Ethiopia | CS | Hospitals only | All HCWs | Random | All others standard precautions | 162 | 282 | 0.5744 | 0.0294 | 7 |
| Hussen SH et al. [30] | 2017 | Ethiopia | CS | Hospitals only | All HCWs | Non-probability | All others standard precautions | 164 | 271 | 0.6051 | 0.0296 | 7 |
| Temesgen C et al. [31] | 2014 | Ethiopia | hybrid | Hospitals only | All HCWs | Non-probability | All others standard precautions | 206 | 326 | 0.6319 | 0.0267 | 7 |

*(Continued)*

**Table 1.** (Continued)

| Study Name | Year | Country | design | Study setting | Profession | Sampling | Types of IPC practices assessed | Event | sample size | proportion | seproportion | Quality |
|---|---|---|---|---|---|---|---|---|---|---|---|---|
| Iliyasu G et al. [32] | 2016 | Nigeria | CS | Hospitals only | All HCWs | Non-probability | Hand hygiene only | 130 | 200 | 0.6500 | 0.0337 | 7 |
| Laraqui O et al. [33] | 2008 | Morocco | transversal survey | All HCFs | All HCWs | Non-probability | All others standard precautions | 1368 | 2086 | 0.6558 | 0.0104 | 8 |
| Sahiledengle B et al. [34] | 2018 | Ethiopia | CS | All HCFs | All HCWs | Non-probability | All others standard precautions | 400 | 605 | 0.6611 | 0.0192 | 8 |
| Paudyal P et al. [35] | 2008 | Nepal | CS | Hospitals only | All HCWs | Non-probability | Hand hygiene only | 227 | 324 | 0.7006 | 0.0254 | 7 |
| Askarian M et al. [13] | 2005 | Iran | CS | Hospitals only | Nurses only | Non-probability | All others standard precautions | 53 | 270 | 0.1962 | 0.0241 | 7 |
| Sarani H et al. [36] | 2015 | Iran | CS | Hospitals only | Nurses only | Random | All others standard precautions | 71 | 170 | 0.4176 | 0.0378 | 7 |
| Woldegioris T et al. [37] | 2019 | Ethiopia | CS | Hospitals only | Nurses only | Non-probability | All others standard precautions | 92 | 204 | 0.4509 | 0.0348 | 7 |
| Mengesha A et al. [38] | 2020 | Ethiopia | CS | Hospitals only | Nurses only | Non-probability | All others standard precautions | 200 | 409 | 0.4889 | 0.0247 | 8 |
| Sahiledengle B. [39] | 2019 | Ethiopia | CS | Hospitals only | Nurses only | Random | All others standard precautions | 134 | 273 | 0.4908 | 0.0302 | 7 |
| Bekele I et al. [40] | 2018 | Ethiopia | CS | Hospitals only | Nurses only | Non-probability | All others standard precautions | 148 | 231 | 0.6406 | 0.0315 | 7 |
| Suliman M et al. [41] | 2018 | Jordan | CS | Hospitals only | Nurses only | Non-probability | All others standard precautions | 161 | 247 | 0.6518 | 0.0303 | 7 |
| Mursy SMM et al. [42] | 2019 | Sudan | CS | Hospitals only | Nurses only | Non-probability | All others standard precautions | 72 | 110 | 0.6545 | 0.0453 | 7 |
| Kebebe B et al. [43] | 2015 | Ethiopia | CS | Hospitals only | All HCWs | Non-probability | Hand hygiene only | 13 | 17 | 0.7647 | 0.1028 | 7 |
| Legese T et al. [44] | 2015 | Ethiopia | CS | Hospitals only | All HCWs | Non-probability | Hand hygiene only | 72 | 73 | 0.9863 | 0.0136 | 7 |
| Alemu B et al. [45] | 2015 | Ethiopia | CS | Hospitals only | All HCWs | Non-probability | Hand hygiene only | 32 | 47 | 0.6808 | 0.0679 | 7 |
| Zewde GT [46] | 2019 | Ethiopia | CS | Hospitals only | All HCWs | Random | Hand hygiene only | 93 | 125 | 0.7440 | 0.0390 | 7 |
| Koech SJ et al. [47] | 2021 | Keniya | CS | Hospitals only | All HCWs | Random | Hand hygiene only | 150 | 301 | 0.4983 | 0.0288 | 7 |
| Ekwere T et al. [48] | 2013 | Nigeria | CS | Hospitals only | All HCWs | Random | Hand hygiene only | 301 | 430 | 0.7000 | 0.0220 | 7 |
| Gajida Au et al. [49] | 2020 | Nigeria | CS | Hospitals only | All HCWs | Non-probability | Hand hygiene only | 236 | 302 | 0.7814 | 0.0237 | 7 |
| Oluwagbemiga AO et al. [50] | 2020 | Nigeria | CS | All HCFs | All HCWs | Random | Hand hygiene only | 70 | 137 | 0.5109 | 0.0427 | 7 |
| Alemayehu R et al. [51] | 2016 | Ethiopia | CS | Hospitals only | All HCWs | Random | Hand hygiene only | 205 | 208 | 0.9855 | 0.0082 | 7 |

(*Continued*)

**Table 1.** (Continued)

| Study Name | Year | Country | design | Study setting | Profession | Sampling | Types of IPC practices assessed | Event | sample size | proportion | seproportion | Quality |
|---|---|---|---|---|---|---|---|---|---|---|---|---|
| Gezie H et al. [52] | 2019 | Ethiopia | CS | Hospitals only | All HCWs | Random | Hand hygiene only | 56 | 191 | 0.2931 | 0.0329 | 7 |
| Jemal S [53] | 2018 | Ethiopia | CS | Hospitals only | All HCWs | Random | Hand hygiene only | 39 | 91 | 0.4285 | 0.0518 | 7 |
| Joshi SK et al. [54] | 2013 | Nepal | CS | Hospitals only | All HCWs | Non-probability | Hand hygiene only | 302 | 336 | 0.8988 | 0.0164 | 7 |
| Yallew WW et al. [55] | 2015 | Ethiopia | CS | Hospitals only | All HCWs | Non-probability | All others standard precautions | 232 | 422 | 0.5497 | 0.0242 | 7 |
| Asmr Y et al. [56] | 2019 | Ethiopia | CS | Hospitals only | All HCWs | Non-probability | All others standard precautions | 77 | 128 | 0.6015 | 0.0432 | 7 |
| Abreha N [57] | 2018 | Ethiopia | CS | Hospitals only | Nurses only | Non-probability | All others standard precautions | 78 | 108 | 0.7222 | 0.0430 | 7 |

Abbreviations: SC—cross-sectional, HCWs–Healthcare workers

from 6.3% to 98.6%; the smallest was in Nigeria for post-exposure prophylaxis, while the largest was in Ethiopia for hand hygiene (Table 1).

## Meta-analysis for IPC practices among healthcare workers

The pooled prevalence of standard precaution practices among healthcare workers in LMICs was 53%, with a 95% confidence interval of 47–59. With I2 values of 98.79%, a heterogeneity chi square (Q) of 7730.12, an estimate of the study variance (Tau-Squared) of 902.09, and a P-value less than 0.001, we used a random effect meta-analysis model due to this high heterogeneity (Fig 2). The Galbraith plot reports standardized effect size against precision. The Galbraith plot revealed a large number of studies that were outside the 95% confidence limits (+/-2) that indicate heterogeneity. This corresponded to the I-squared number (Fig 3). The prevalence study's high I2 values, however, may not be an absolute indicator of the existence of observed variation because their estimation can be influenced by other variables such as the number of studies and pooled estimates [56]. In prevalence studies, the I-squared value could be large as the number of studies increased and the pooled prevalence became between 10% and 90%.

## Sensitivity analysis

The sensitivity analysis was investigated using the metaninf syntax by the proportion and its 95% confidence interval. The meta-analysis recalculates the estimate and confidence interval by leave-out-one study at a time. The table and figure of the results showed that each individual study had no influence on the overall meta-analysis summary estimate because all "omitted" analysis estimates lied within the confidence interval of the "combined" analysis estimate (Table 2, Fig 4).

The graph visualized the same result with the table that no individual study was suspected to excessive influence. As shown on Fig 4, there was no estimate of individual study "omitted" analysis lied outside the confidence interval of the "combined" analysis.

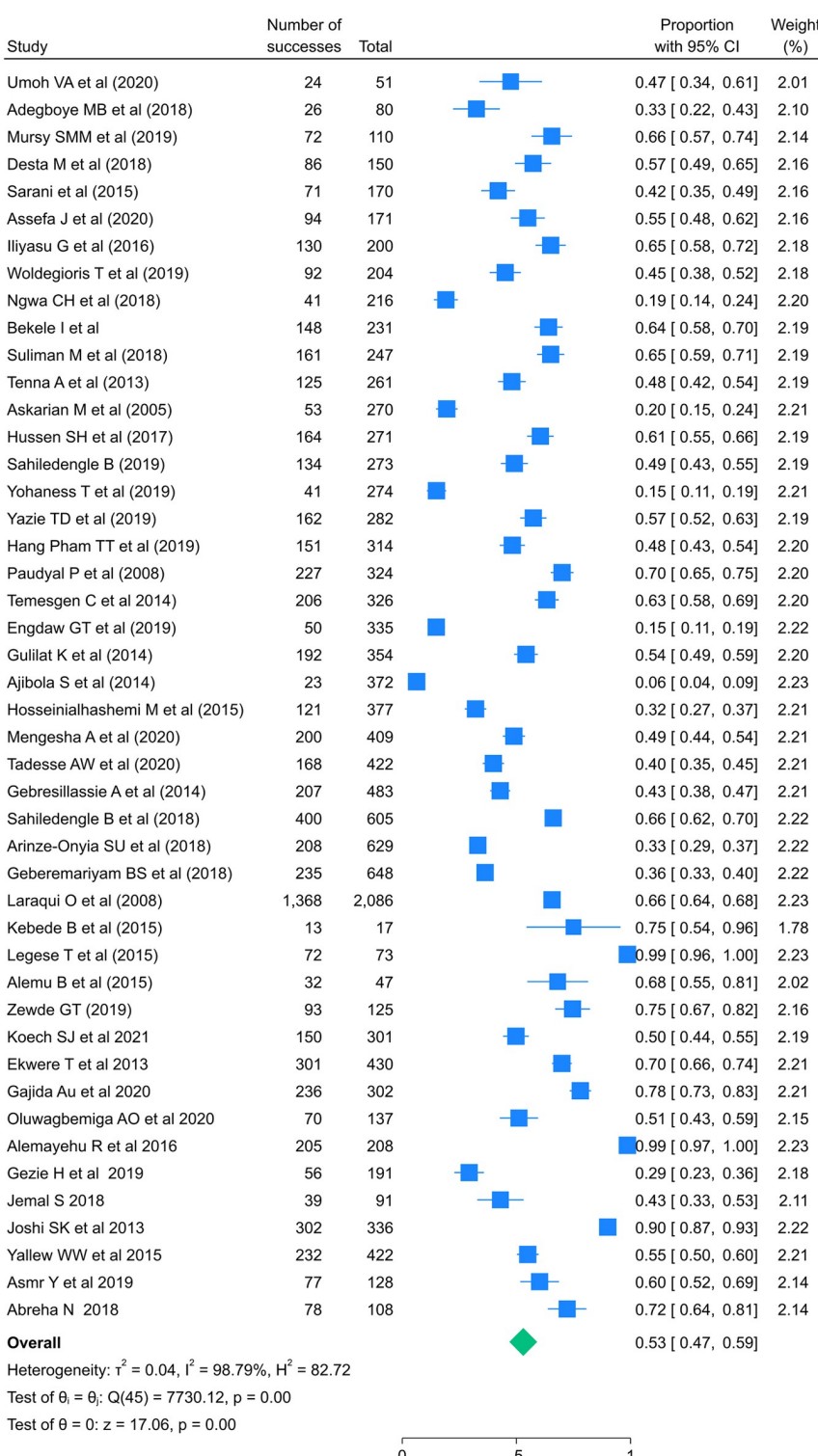

**Fig 2. Forest plot of the pooled prevalence of standard precautions practice among HCWs in LMICs.**

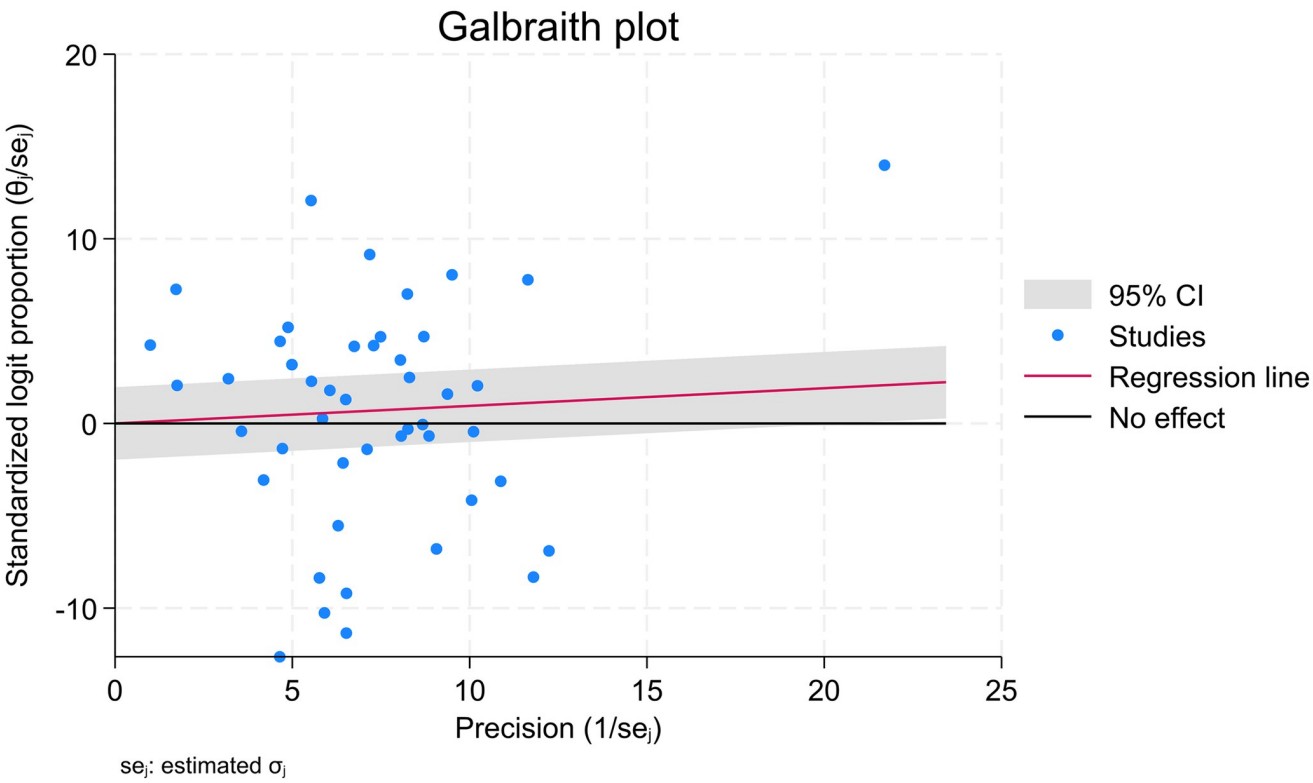

**Fig 3. Galbraith plot of proportion\*precision / standard error against precision (1/standard error) among HCWs in LMICs.**

### Subgroup analysis

The subgroup analyses were conducted by publication years, types of IPC practiced assessed, study setting, profession, and sampling techniques. The pooled prevalence of standard Precautions practices was lesser 51 (95%CI: 46, 58) after publication year of 2015 than before 56% (95% CI: 44, 68). The pooled prevalence of standard Precautions practices for all type of standard precautions were 49% (95%CI: 42, 55) while it is 59% (95%CI: 48, 70) for hand hygiene only practice (Figs 5 and 6 and Table 3).

### Publication bias

There are arguments about assessing publication bias in proportion meta-analysis. As publication bias was assumed in the context of comparative studies that studies with positive results are more frequently published than studies with negative results, this doesn't make sense for proportion studies with no comparison [58, 59]. As a result, studies came up with various suggestions that Barker et al. do not suggest using funnel plots, Egger's and Begg's tests for proportional meta-analysis [58], Wang N. suggests interpreting these tests with caution [59], and Hunter et al. suggest using study size against log odds to assess publication bias in proportion meta-analysis [60]. After careful consideration of this evidence, we decided to use the funnel plot and Egger's test at least to visualize the distribution of studies and statistical testing along the pooled result. The funnel plot displayed that many points representing each study are scattered outside the inverted funnel. As a result, we conducted a trim and fill analysis, and the results showed there were missed studies on the left side of the overall effect size. The Egger's

**Table 2. Sensitivity analysis of standard precautions practices among HCWs in LMICs.**

| Study omitted | Estimate | 95% confidence interval | |
|---|---|---|---|
| Ajibola S et al | 0.53183764 | 0.44342586 | 0.62024945 |
| Engdaw GT et al | 0.53505021 | 0.44666272 | 0.6234377 |
| Yohaness T et al | 0.52782702 | 0.43922654 | 0.61642754 |
| Ngwa CH et al | 0.52961248 | 0.44092992 | 0.61829507 |
| Hosseinialhashemi M et al | 0.53306305 | 0.44445473 | 0.62167144 |
| Adegboye MB et al | 0.53013498 | 0.44141456 | 0.61885536 |
| Arinze-Onyia SU et al | 0.52790254 | 0.43908563 | 0.61671942 |
| Geberemariyam BS et al | 0.53232896 | 0.4436228 | 0.6210351 |
| Tadesse AW et al | 0.53816926 | 0.45038915 | 0.62594944 |
| Gebresillassie A et al | 0.52810663 | 0.43921769 | 0.61699557 |
| Umoh VA et al | 0.52785665 | 0.4389295 | 0.6167838 |
| Tenna A et al | 0.53171229 | 0.4428663 | 0.62055832 |
| Hang Pham TT et al | 0.53803438 | 0.4503389 | 0.62572986 |
| Gulilat K et al | 0.52889729 | 0.43992433 | 0.61787021 |
| Assefa J et al | 0.53144741 | 0.44256306 | 0.62033176 |
| Desta M et al | 0.53908437 | 0.45209241 | 0.62607628 |
| Yazie TD et al | 0.52958232 | 0.44060019 | 0.61856443 |
| Hussen SH et al | 0.53167158 | 0.44272241 | 0.62062073 |
| Temesgen C et al | 0.52675849 | 0.43766403 | 0.61585295 |
| Iliyasu G et al | 0.52829784 | 0.43918979 | 0.61740589 |
| Laraqui O et al | 0.53909802 | 0.45245808 | 0.62573797 |
| Sahiledengle B et al | 0.5302996 | 0.44118759 | 0.61941165 |
| Paudyal P et al | 0.54102051 | 0.46157533 | 0.6204657 |
| Askarian M et al | 0.53525162 | 0.44670743 | 0.62379575 |
| Sarani H et al | 0.53149444 | 0.44235042 | 0.62063849 |
| Woldegioris T et al | 0.53352815 | 0.44460645 | 0.62244982 |
| Mengesha A et al | 0.53284937 | 0.44373438 | 0.62196434 |
| Sahiledengle B | 0.52763981 | 0.43788964 | 0.61739004 |
| Bekele I et al | 0.53504688 | 0.44632494 | 0.62376875 |
| Suliman M et al | 0.5343315 | 0.44534662 | 0.62331641 |
| Mursy SMM et al | 0.52778721 | 0.43478063 | 0.62079376 |
| Kebebe B et al | 0.52584308 | 0.43754247 | 0.61414373 |
| Legese T et al | 0.52026844 | 0.43436864 | 0.60616821 |
| Alemu B et al | 0.52733737 | 0.43890855 | 0.61576623 |
| Zewde GT | 0.52582884 | 0.43720597 | 0.61445171 |
| Koech SJ et al | 0.53128153 | 0.44233251 | 0.62023062 |
| Ekwere T et al | 0.52676839 | 0.43743423 | 0.61610252 |
| Gajida Au et al | 0.52494293 | 0.4359903 | 0.61389554 |
| Oluwagbemiga AO et al | 0.53099173 | 0.44236633 | 0.61961722 |
| Alemayehu R et al | 0.52012235 | 0.44266844 | 0.5975762 |
| Gezie H et al | 0.53584146 | 0.44746545 | 0.62421751 |
| Jemal S | 0.53278822 | 0.44430393 | 0.62127256 |
| Joshi SK et al | 0.52227837 | 0.43403324 | 0.6105234 |
| Yallew WW et al | 0.53013545 | 0.44086912 | 0.61940181 |
| Asmr Y et al | 0.52899146 | 0.44035307 | 0.61762983 |
| Abreha N | 0.52632678 | 0.43773696 | 0.61491656 |
| Combined | 0.53055564 | 0.44317112 | 0.61794015 |

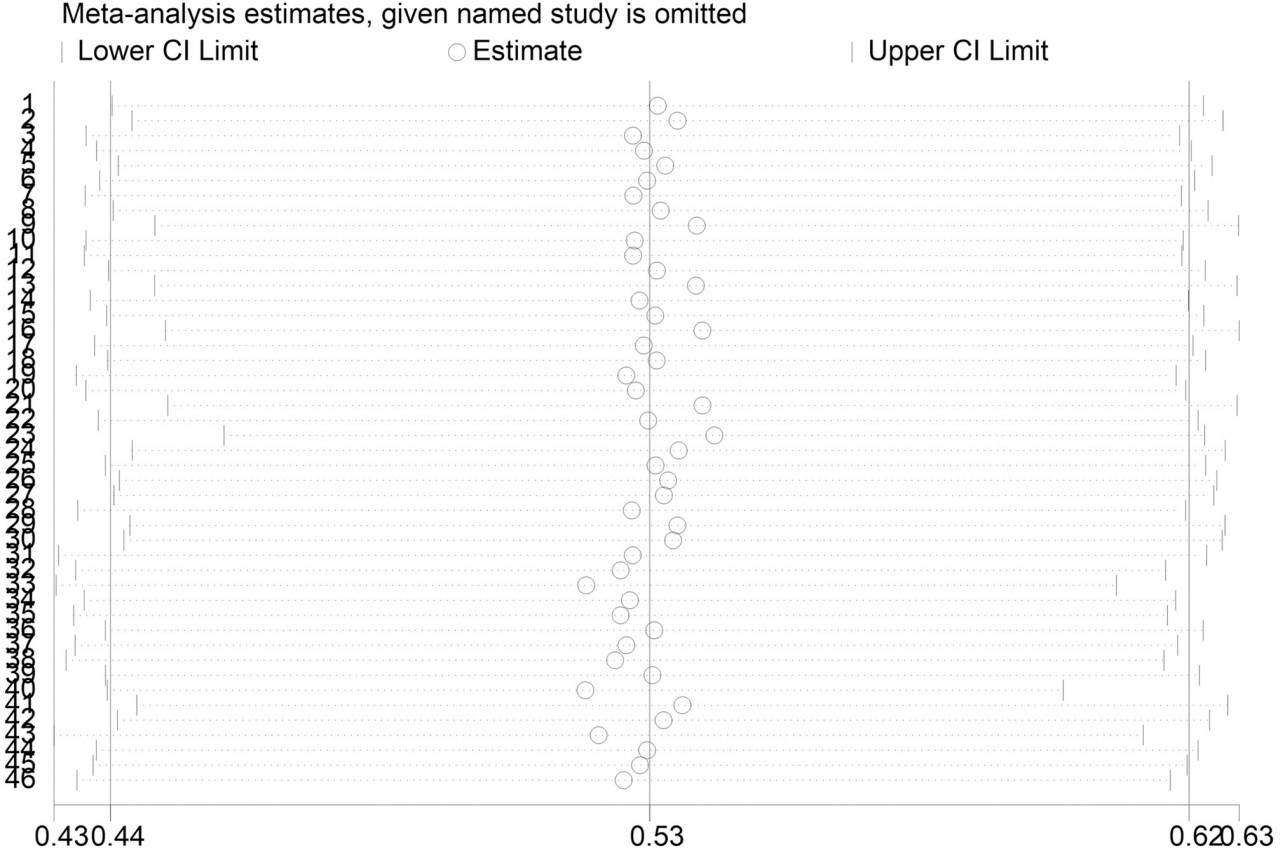

**Fig 4. The graph visualize the influence of each individual study on the combined result in prevalence of standard precautions practices in LMICs.**

tests also showed the evidence of significant missing of studies with P-value of 0.001 (Figs 7–9).

## Discussion

According to the CDC document [1], standard precautions include hand hygiene, use of PPE, respiratory hygiene, sharps safety, safe injection practices, sterile medical instruments and devices, and clean and disinfected environmental surfaces. Hence, articles that assessed one or more of these standard precaution components were included in this systematic review and meta-analysis. This study was conducted to synthesize evidence and estimate the pooled prevalence of standard precautions among healthcare workers in LMICs. According to our search, the distribution of the identified articles varied across LMICs. The majority of the articles were identified in Ethiopia. The possible explanation for this variation might be due to the interest in research focus areas and the difference in magnitude of the problem. The other possible reason may be due to differences in the publication of articles and their availability in online databases. This information can inform researchers to fill the research gap concerning standard precautions in countries with little or no research.

The finding indicated that the pooled prevalence of standard precautions was suboptimal in LMICs with respect to the WHO recommendation that all healthcare workers in all patient care settings should consistently implement them [5]. This is consistent with the pooled

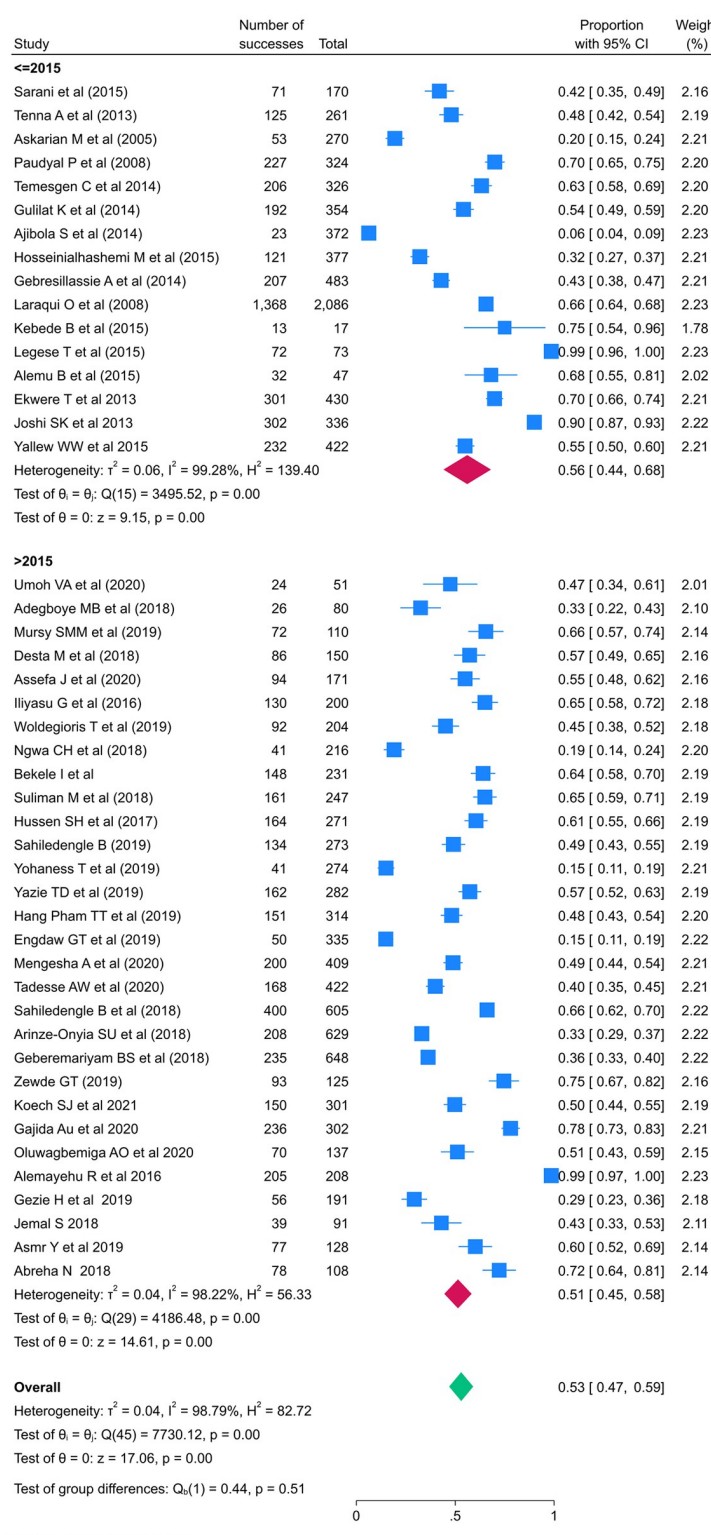

**Fig 5. Forest plot of the pooled prevalence of standard precautions practices sub grouped by publication year among HCWs in LMICs.**

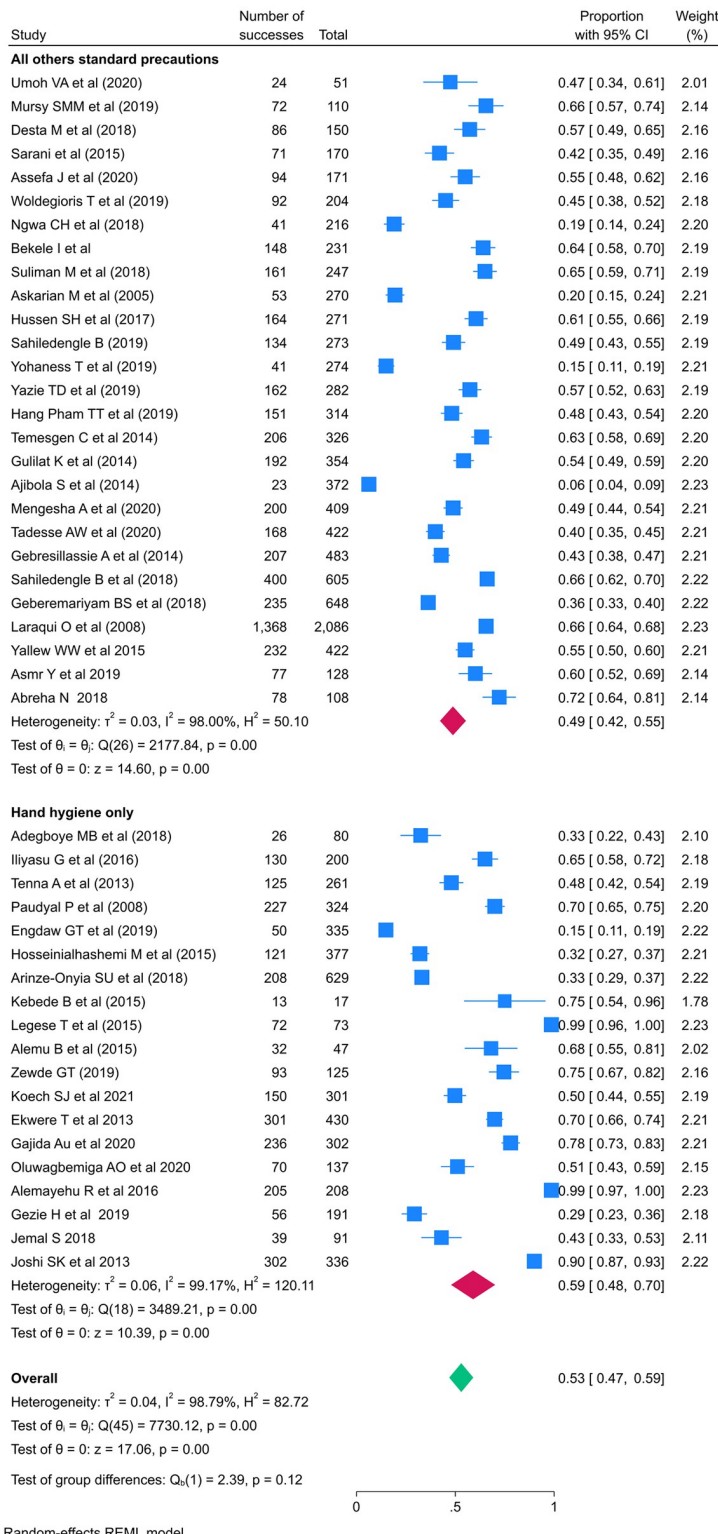

**Fig 6. Forest plot of the pooled prevalence of standard precautions practices sub grouped by practice type assessed among HCWs in LMICs.**

**Table 3. Sub- group analysis for other characteristics of standard precautions practices among HCWs in LMICs.**

| Sub-groups | Characteristics | Studies | Prevalence(95% CI) | Heterogeneity Statistics | | |
|---|---|---|---|---|---|---|
| | | | | Heterogeneity | P-value | $I^2$ |
| Study setting | Only hospitals all type | 38 | 53 (46, 60) | 7448.38 | <0.001 | 98.88 |
| | Hospitals and others | 8 | 52 (45, 60) | 263.04 | <0.001 | 95.89 |
| Profession | Nurse only | 9 | 52 (42, 63) | 246.85 | <0.001 | 96.21 |
| | All other types of HCWs | 37 | 53(46, 60) | 7356.10 | <0.001 | 99.02 |
| Sampling technique | Random | 19 | 45(35, 56) | 5502.37 | <0.001 | 98.98 |
| | Non-probability | 27 | 58(51, 65) | 2157.05 | <0.001 | 98.27 |

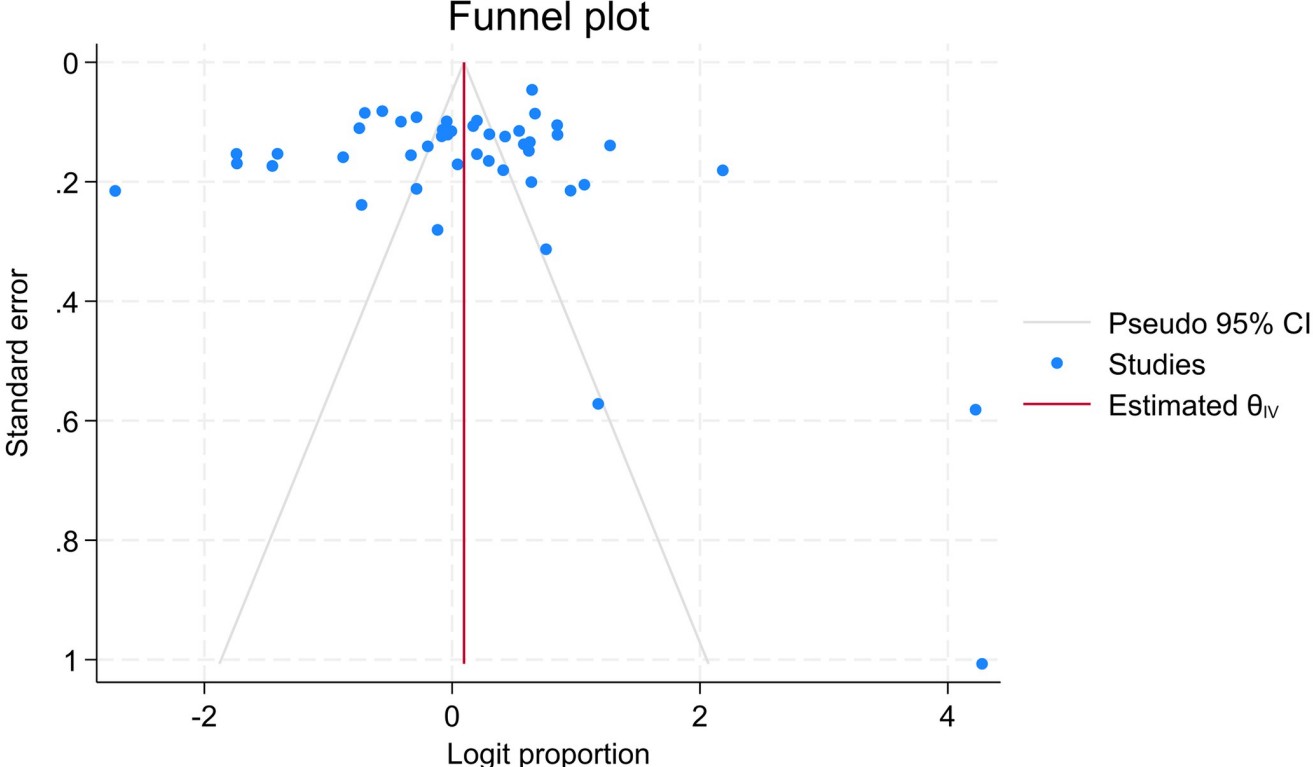

**Fig 7. Funnel plot of standard precautions practices among HCWs in LMICs.**

prevalence of safe infection prevention practices in Ethiopia from 10 studies [61]. Another systematic review conducted on nurses' adherence to infection control and prevention practices indicated average to poor nursing practices [62]. This finding can provide relevant information for healthcare workers and decision-makers to act on this important public health issue. As per the WHO recommendation, healthcare workers should consistently adhere to the practices to reduce HAIS and prevent the spread of these infections. HAIs are infections that patients acquire while receiving healthcare service or treatment for medical or surgical conditions [63]. The HAIs remain persistent in LMICs by exerting a burden on healthcare facilities, healthcare workers, patients, and their families, as well as the governments. The HAIs can cause prolonged hospital stays, massive financial losses, and increased microbial resistance

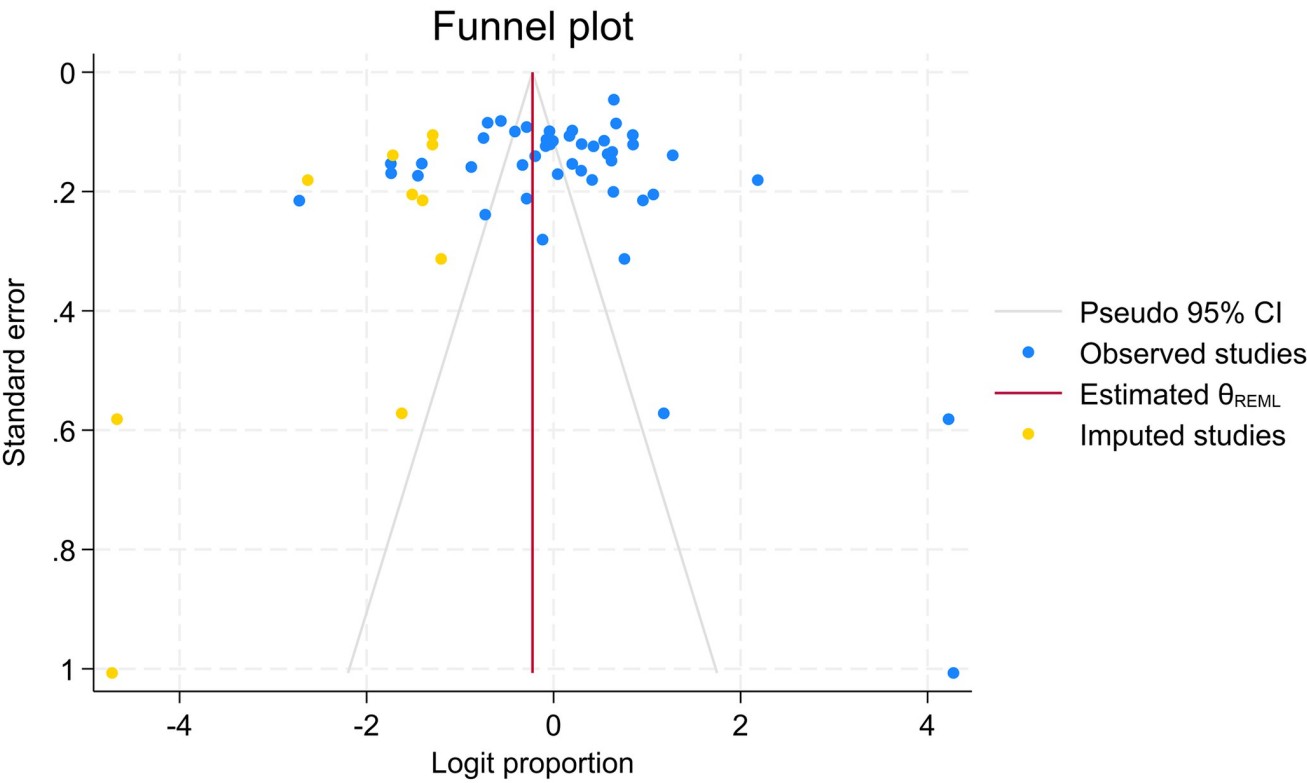

**Fig 8. Funnel plot with trim and fill of standard precautions practices among HCWs in LMICs.**

Regression-based Egger test for small-study effects

Random-effects model

Method: REML

H0: beta1 = 0; no small-study effects

$$\text{beta1} = \quad 4.51$$

$$\text{SE of beta1} = \quad 1.132$$

$$z = \quad 3.98$$

$$\text{Prob} > |z| = \quad 0.0001$$

**Fig 9. Egger's test for standard precautions practices among HCWs in LMICs.**

[63]. So, the pooled prevalence of safe standard precautions in this study can be considered very low, whereas the problem remains high.

In this review close to half of the articles assessed about hand hygiene practice, while components of standard precautions are around seven [1], and the largest majority of studies were conducted in hospitals only. The distribution of studies across countries in the LMICs also varied, from no study to more than half studies. These can inform researchers and decision-makers about evidence gaps from the perspective of practice types, health facility types, and countries. The subgroup pooled prevalence of hand hygiene was higher than the overall pooled prevalence. The possible reasons for this difference may be that most studies were conducted on hand hygiene practice, which may also increase robustness of evidence.

Generally, the result of this study can be important to inform healthcare workers, planners, decision-makers, and researchers to highlight the magnitude of the problem and identify areas where evidence lacks from various perspectives. Based on the findings, healthcare workers should optimize their practices to meet the WHO recommendations on standard precaution. Decision-makers need to understand the magnitude of the problems and the evidence gaps to enhance the implementation of standard precautions. Researchers should focus their research on health facility types and standard precaution components that have been little studied.

## Limitation of the study

The use of purposive inclusion criteria, language of included studies, and a few access for databases were the limitation of this review. As only English-language articles were included in this study, there was obvious systematic bias that missed articles published in other languages. We used the PubMed and Google Scholar databases as the sources of articles for this study. However, as a result of not including more traditional academic databases, selection bias by geographical region, the missing of relevant studies, heterogeneity due to variation in population characteristics, and a lack of standardization may cause less representativeness, generalizability, and quality of included studies as different databases have different criteria to index articles. The variation in the number of included study across the LMICs may create non-representativeness and the result may need caution in interpretation and application. The studies included in this meta-analysis were observational studies and self-reported. So this may cause over estimation of pooled estimates.

## Conclusions

The pooled prevalence of standard precautions practices among healthcare workers in LMICs was suboptimal. Evidence gaps were identified from the perspective of facility types, practice types, and countries. The findings of this study can have substantial implications for healthcare practices, policymaking and research by providing robust evidence with synthesized and pooled data from multiple studies.

## Supporting information

**S1 Table. PRISMA checklist.**
(PDF)

**S2 Table. Systematic review and meta-analysis extracted data.**
(PDF)

## Author Contributions

**Conceptualization:** Mengistu Yilma.

**Formal analysis:** Mengistu Yilma.

**Validation:** Girma Taye, Workeabeba Abebe.

**Writing – original draft:** Mengistu Yilma.

**Writing – review & editing:** Girma Taye, Workeabeba Abebe.

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
