## [Decision Letter · Decision Letter 0]

11 Dec 2023

PONE-D-23-16216Magnitude of Standard Precautions Practices among Healthcare Workers in Health Facilities of Low and Middle Income Countries: A Systematic Review and Meta-analysis.PLOS ONE

Dear Mengistu Yilma,

Thank you for submitting your manuscript to PLOS ONE. After careful consideration, we feel that it has merit but does not fully meet PLOS ONE’s publication criteria as it currently stands. Therefore, we invite you to submit a revised version of the manuscript that addresses the points raised during the review process. Below, I have summarised the key comments by the reviewers that must be addressed. Any additional comment here must be addressed, in addition to those stated by the reviewers.

**Abstract**

Consider merging lines 12-14 as both appear to say the same thing. Provide a succinct context for the study by including burden and impact of healthcare-associated infections and the challenges of implementing standard precautions in LMICs.The methods section should briefly describe the inclusion criteria, and statistical methods used for the meta-analysis.The conclusion section should highlight the implications and significance of the findings. It should avoid repeating the results.Report on the confidence intervals of the pooled prevalence in the abstract.

** Introduction**

Please describe the rationale for the review in the context of existing knowledge. This may involve summarising key studies carried out in the area and establish gap in evidence in support of the current study. Refer to PRISMA checklist point 3.

**Methods**

Please address this comment by reviewer 1 ‘the authors' approach to searching for articles leaves room for evidence/article selection bias. Additional searches for both published and grey literature should have been done. Mention Google scholar as part of the sites searched for additional literature’. I would like to add that if Google scholar was searched, then that becomes your source of grey literature search.Line 86-88, the authors wrote ‘We systematically searched articles from PubMed Central, EMBASE, MEDLINE, Global Health, African Journals Online (AJOL), and Google Scholar. However, the Figure (PRISMA) shows search results for only PubMed. Please clarify this inconsistency.From line 88 – the authors discussed study selection and management of data retrieved. This section may be the wrong section to discuss managing retrieved data as no search has been done. I suggest having a separate heading after the search strategy and writing out how the total retrieved search results was managed.Please clarify whether reviewers who screened each record worked independently. The authors mentioned that disagreements were resolved by a third author. Clarify this process – was it a discussion, a consensus meeting, or another method?Lines 166-161 -More details on the data extraction process would be beneficial. For instance, explain if a standardized form was used for data extraction and how discrepancies between extractors were resolved.Please comment on the criteria and justification for conducting the subgroup analysis by publication year, types of IPC practices, study setting, profession, and sampling technique, and the significance and relevance of the findings from each subgroup.Line 150: It’s not clear what you mean by poor quality score.

**Discussion**

The authors did not report how the review findings compare with other relevant evidence from previous reviews or primary studies. Please refer to  PRISMA guideline 23a

Please submit your revised manuscript by **22nd January, 2024** . If you will need more time than this to complete your revisions, please reply to this message or contact the journal office at plosone@plos.org. Please include the following items when submitting your revised manuscript:

We look forward to receiving your revised manuscript.

Kind regards,

Isaac Amankwaa, Ph.D.

Guest Editor

PLOS ONE

Journal Requirements:

https://aricjournal.biomedcentral.com/articles/10.1186/s13756-020-00801-x

https://journals.plos.org/plosone/article?id=10.1371%2Fjournal.pone.0245469

https://linkinghub.elsevier.com/retrieve/pii/S2213398420302104

4. In your revision ensure you cite all your sources (including your own works), and quote or rephrase any duplicated text outside the methods section. Further consideration is dependent on these concerns being addressed.

Reviewers' comments:

Reviewer's Responses to Questions

**Comments to the Author**

1. Is the manuscript technically sound, and do the data support the conclusions?

Reviewer #1: Yes

Reviewer #2: Yes

2. Has the statistical analysis been performed appropriately and rigorously? 

Reviewer #1: Yes

Reviewer #2: Yes

3. Have the authors made all data underlying the findings in their manuscript fully available?

Reviewer #1: Yes

Reviewer #2: Yes

4. Is the manuscript presented in an intelligible fashion and written in standard English?

Reviewer #1: Yes

Reviewer #2: Yes

5. Review Comments to the Author

Reviewer #1: Thank you for the opportunity to review this manuscript. Generally, the problem and the justification for this study is not well articulated in the background. The methods that guided the conduct of the systematic review is not reportedwith sufficient detail and transparency. My main comments are appended below.

Abstract:

Introduction- the introduction in the abstract should capture the problem and the gap identified in the literature. You may delete some of the general background information already provided with the problem and the gap.

Results: The results should also capture some of the main findings of the study (in accordance with the objectives of this review).

Main manuscript.

Introduction-

1. Please use a box bracket instead to enclose the intext citation.

2. The problem is not well stated. The authors need to further look at the introduction again. The introduction should clearly establish the problem, context and significance.

3. Line 76:........ The paucity of evidence about the current subject............ this statement is incorrect and only shows you did not do a good review of the literature. the outbreak of COVID-19, Ebola, Flu has drawn more attention to infection prevention practices leading to increased research and publication in this area. In 2023 alone, hundreds of articles have been published various components of infection prevention.

4. In the concluding part of your revised introduction, it should capture the implication of your study on healthcare policies.

Methods

5. Line 88: Google scholar is not considered as a major database.

6. the authors' approach to searching for articles leaves room for evidence/article selection bias. Additional searches for both published and grey literature should have been done. Mention Google scholar as part of the sites searched for additional literature.

7. Reference list of included articles should have been used as snowball. Hand searches and consultation with experts /personal correspondence should have been mentioned.

8. How many reviewers conducted the electronic search for articles and on which day did you begin and when was the final searches completed?

9. I stand to be corrected but i doubt if you adhered to the reported items outlined in PRISMA.

10. How were duplicates removed?

Search Strategy

11. Line 105-128: I doubt if PubMed will understand the search language and give you accurate and precise search outcome. When two words are combined together as infection prevention or infection control, quotation marks are used ("infection prevention") to communicate to PubMed that it is one word.

Inclusion

12. Be specific with the study design type. you mentioned observational studies, were case studies and qualitative research articles included? Please clarify.

13. the publication year should be indicated in the inclusion : for instance, articles published within the 10 years from 2013 to 2023.

14. Line 150: Its not clear what you mean by poor quality score.

15. line 157: Statement is unclear.

Results

16. Line 183-188: Its unthinkable and unrealistic how 112 articles were retrieved from EMBASE, MEDLINE, Global Health, AJOL and Google scholar. I can emphatically say you only searched PubMed and Google scholar.

17. Page 12: Provide a legend for the table and state the abbreviations in full.

18. Fig 1: PRISMA flow diagram: it is unclear how the authors arrived at the 429 articles from the 8016 remaining articles.

19. Discuss the limitations with the few databases used, purposive inclusion criteria used and the limits placed on language of published articles.

Reviewer #2: Many thanks for the opportunity to review this manuscript. I would like to thank the authors for the time and effort that went into the preparation of the manuscript. I have a few comments for the authors;

1. Line 54-line 58--I believe these two statements could be put together into a single concise statement. At the moment they appear unwieldy and have a few repetitions.

2. Line 64-line 65--"In recent years, more attention has been given to IPC practices in health care facilities as a result of the occurrence of new and emerging health problems". This is a claim that was not supported. What is the evidence that more attention has been given to IPC practices in recent years?

3. Line 67-line 70--According to the report from the Organization for Economic Co-operation and Development (OECD), it is

estimated that promoting simple IPC measures such as hand hygiene could reduce the AMR burden by about 40%.

What OECD report is this? The authors need to provide a citation for this report

4. Line 72-75--Healthcare associated infections are the major concern worldwide for the health systems.

This statement sounds as if this is the only 'major' issue plaguing health care systems. However, this is not necessarily true. What about the rising incidence of non-communicable diseases in LMICs? What about the challenges associated with Universal Health Coverage? It is probably fair to say it is AMONG the issues but not THE MAJOR issue. The same applies to the statement that follows--- "Ebola, MERS, COVID-19, and monkey pox are the major concerns that have imposed an additional burden on the health system in recent years".

While these are among the plethora of challenges that affect healthcare systems, they are not the only ones.

5. In line 75-76, the authors wrote "In addition to the aforementioned situations, the

paucity of evidence about the current subject drove the initiation of this meta-analysis"----The problem is not well established here. What shows that there is a paucity of research on this issue? The authors should have teased out the problem in a better way. For example, there may be single studies in LMICs but these are scattered and not pulled together for evidence purposes etc. I feel the problem or question should be framed in such a way that it will inform the need for a systematic review in this area. As it stands, we only have a piecemeal presentation of the main research issue and why it is worth using a systematic review.

6. Discussion section--In general, the discussion section is very lame. What do these findings mean in the light of infection control in LMICs? What are the implications for healthcare practice, research and policy? A great number of studies were conducted in Nigeria—how does this impact your interpretation (although the authors talked briefly about Ethiopia)? The authors should use published research to explain and discuss the trends in prevalence of infections and how this should inform the evidence scenario for HAIs in low resource settings moving forward.

7. There are so many repetitions in the manuscript. For example, under the subsection "Meta-analysis for IPC practices among healthcare workers"--there are so many repetitions there. The authors should check for other statements that are repeated throughout the manuscript. e.g. This systematic review and meta-analysis included 46 papers in total.

8. The article should be edited for grammatical errors. There are so many editorial mistakes in the manuscript. The authors should take some time to read through the document critically and get them corrected.

9. The first statement in the discussion section refers to a CDC document. What document is this? A citation is needed there

10. The conclusion section should present a succinct summary of the findings as well as the key lessons, implications or recommendations regarding how this evidence informs infection control practices and policies for healthcare workers, patients etc

6. PLOS authors have the option to publish the peer review history of their article (what does this mean?). If published, this will include your full peer review and any attached files.

Reviewer #1: No

Reviewer #2: No

---

## [Author Response · Author response to Decision Letter 0]

26 Jan 2024

Dear editor

I would like to thank you very much for the coordination and summary of important comments from the reviewers. We have tried to incorporate all comments and made further improvements to our paper based on the valuable comments.

Dear reviewers 

First of all, I would like to express my appreciation and gratitude for your valuable comments that significantly improve the quality of our research and for taking the time to review thoroughly. We have tried to make the necessary amendments and additional improvements to the research accordingly. we have uploaded a line-by-line responses for all your comments with separate document.

---

## [Decision Letter · Decision Letter 1]

15 Feb 2024

PONE-D-23-16216R1Magnitude of Standard Precautions Practices among Healthcare Workers in Health Facilities of Low and Middle Income Countries: A Systematic Review and Meta-analysis.PLOS ONE

Dear Dr. Dilnessie,

Thank you for submitting your manuscript to PLOS ONE. First and foremost, I want to express my appreciation for the revisions made. Your work contributes significantly to the field, and we commend your efforts.

However, upon careful examination, there are a few points that require further clarification and attention to ensure transparency, rigor, and accurate reporting. I would like to highlight these aspects:

**PubMed Database Screening and Rayyan Process:**
In your manuscript, you mentioned that 317 articles were selected from PubMed Central based on your search strategy. However, the PRISMA flow chart indicates an initial identification of 8,115 records through PubMed database screening.Could you kindly elaborate on the primary screening process on PubMed pages? Specifically, why was this step performed before using Rayyan for further selection?Additionally, how were the 211 possible duplicates identified by the Rayyan system resolved? Were they excluded during the PubMed screening or at a later stage?
**“Other Sources” and Database Clarity:**
From the PRISMA Flow Chart, you state that PubMed was the sole database used in this systematic review.However, you also mention retrieving 112 studies from other sources. It remains unclear whether these “other sources” refer to databases such as EMBASE, MEDLINE, Global Health, African Journals Online (AJOL), and Google Scholar.To enhance transparency, could you explicitly clarify which databases fall under the category of “other sources”? If these databases include EMBASE, MEDLINE, and others, it would be essential to explain why this selection was made.
**Statistical Analysis Review:**
We engaged a statistician to review your statistical analysis. While your work is commendable, there are concerns raised by the reviewer.We kindly request that you address these concerns in your manuscript to ensure the robustness of your findings.

Given the importance of transparency and rigor in systematic reviews, we believe that addressing these points will strengthen your work.

Please submit your revised manuscript by Mar 31 2024 11:59PM. If you will need more time than this to complete your revisions, please reply to this message or contact the journal office at plosone@plos.org. Please include the following items when submitting your revised manuscript:A rebuttal letter that responds to each point raised by the academic editor and reviewer(s). You should upload this letter as a separate file labeled 'Response to Reviewers'.A marked-up copy of your manuscript that highlights changes made to the original version. You should upload this as a separate file labeled 'Revised Manuscript with Track Changes'.An unmarked version of your revised paper without tracked changes. You should upload this as a separate file labeled 'Manuscript'.

We look forward to receiving your revised manuscript.

Kind regards,

Isaac Amankwaa, Ph.D.

Guest Editor

PLOS ONE

Reviewers' comments:

Reviewer's Responses to Questions

**Comments to the Author**

1. If the authors have adequately addressed your comments raised in a previous round of review and you feel that this manuscript is now acceptable for publication, you may indicate that here to bypass the “Comments to the Author” section, enter your conflict of interest statement in the “Confidential to Editor” section, and submit your "Accept" recommendation.

Reviewer #3: (No Response)

2. Is the manuscript technically sound, and do the data support the conclusions?

Reviewer #3: Yes

3. Has the statistical analysis been performed appropriately and rigorously? 

Reviewer #3: N/A

4. Have the authors made all data underlying the findings in their manuscript fully available?

Reviewer #3: Yes

5. Is the manuscript presented in an intelligible fashion and written in standard English?

Reviewer #3: Yes

6. Review Comments to the Author

Reviewer #3: In this paper the authors performed a systematic review and meta-analysis of the prevalence of standard precautions practices among healthcare works in heath facilities of low- and middle-income countries. The final analysis included 46 articles and between study heterogeneity was observed. A pooled prevalence of 53.05% was reported.

I have a few questions/comments specific to the meta-analysis as listed below.

Was a data transformation used in the meta-analysis? As specified below, the values in several plots did not appear to be based on prevalence directly. Maybe a transformation was applied?

Table 1. Add a column of prevalence so that all the information is together.

Table 2. It says log(prevalence) in the column header, but shouldn’t all values be <0 since prevalence cannot be bigger than 1? Why a log-transformation is used?

Table 3. Please give a more detail description of the sample techniques in “others” group. The same comment applies to the “Others” group for Study Design.

Figure 3. What is ”b” in the plot? A transformation of the prevalence?

Figure 4. Again, what is the estimate? A transformation of the prevalence?

Figure 7. Funnel plot: Was Log(prevalence) shown on the x-axis? Shouldn’t all the values be <0 since prevalence is between 0 and 1.

It is worth delving deeper into the causes of heterogeneity. Over half of the studies were in Ethiopia. Does restricting studies to Ethiopia-only alleviate the heterogeneity? Can any factor explain the heterogeneity?

7. PLOS authors have the option to publish the peer review history of their article (what does this mean?). If published, this will include your full peer review and any attached files.

Reviewer #3: No

---

## [Author Response · Author response to Decision Letter 1]

21 Feb 2024

The comments provided by the editor were in 3 categories. 

1. PubMed Database Screening and Rayyan Process: This is a very critical comment, and we made corrections accordingly. We caused confusion for this part while making revisions to the previous comment. This would be difficult if it were published with these errors. Thank you for this major review. 

2. “Other Sources” and Database Clarity: This is also a very pertinent comment that we admitted well and corrected accordingly.

We planned to review databases like EMBASE, MEDLINE, Global Health, African Journals Online (AJOL), and Google Scholar in addition to Pubmed as other sources. This was written in the protocol. However, in the actual review, all 112 articles were retrieved from Google Scholar.

When we wrote the result, we overlooked this issue and remained as it was on the protocol. So now we have corrected it as per your important comments.

Our intention to use Google Scholar was to find articles that were not indexed and could not be found in PubMed.

3. Statistical Analysis Review: we responded line by line to the reviewer's comments and uploaded it as response to reviewer

---

## [Decision Letter · Decision Letter 2]

26 Feb 2024

PONE-D-23-16216R2 NoteMagnitude of Standard Precautions Practices among Healthcare Workers in Health Facilities of Low and Middle Income Countries: A Systematic Review and Meta-analysis.

PLOS ONE

Dear Dr. Dilnessie,

Thank you for submitting your manuscript to PLOS ONE. After careful consideration, we feel that it has merit but does not fully meet PLOS ONE’s publication criteria as it currently stands. Therefore, we invite you to submit a revised version of the manuscript that addresses the points raised during the review process. 

I have reviewed your response to my comment on the databases used for this review. Your response indicates that you relied on only one database. I find this a significant limitation of the systematic review, particularly in terms of the scope of data source Note s and inclusivity criteria. To move this paper to the next level, I will encourage you to address these issues:

**Use of a Single Database:** The current limitations statement acknowledges the use of purposive inclusion criteria and the language of included studies. However, it does not sufficiently address the significant limitation stemming from the utilization of only one database for your literature search. Please expand on this point by discussing the potential impact of this limitation on the comprehensiveness of your review and the generalizability of your findings.**Access to Databases and Language Restrictions: **The brief mention of limited access to databases and the language of included studies warrants further elaboration. It is crucial to explicitly state how these factors might have influenced the selection of studies and the potential biases introduced as a result.Please address the comment by reviewer 3 on the meta-analysis approach used for this review.

Please ensure that each of these points is addressed comprehensively to provide readers with a clear understanding of the scope and potential limitations of your review. This will not only strengthen the credibility of your study but also offer valuable insights for future research in this area.

We look forward to receiving your revised manuscript.

Kind regards,

Isaac Amankwaa, Ph.D.

Guest Editor

PLOS ONE

Reviewers' comments:

Reviewer's Responses to Questions

**Comments to the Author**

1. If the authors have adequately addressed your comments raised in a previous round of review and you feel that this manuscript is now acceptable for publication, you may indicate that here to bypass the “Comments to the Author” section, enter your conflict of interest statement in the “Confidential to Editor” section, and submit your "Accept" recommendation.

Reviewer #3: (No Response)

2. Is the manuscript technically sound, and do the data support the conclusions?

Reviewer #3: (No Response)

3. Has the statistical analysis been performed appropriately and rigorously? 

Reviewer #3: (No Response)

4. Have the authors made all data underlying the findings in their manuscript fully available?

Reviewer #3: (No Response)

5. Is the manuscript presented in an intelligible fashion and written in standard English?

Reviewer #3: (No Response)

6. Review Comments to the Author

Reviewer #3: Is log transformation used throughout the paper including the forest plot, only that the values in the forest plot were back transformed? This needs to be clearly stated in the paper.

I guess my understanding of prevalence is a proportion while percentage (100* proportion) was used throughout the paper. This is fine. However, the standard errors of log Prevalence in Figure 7 seem to be high. Please double check. Please add standard errors of the prevalence in Table 1 as well. If log transformation was used please added the standard errors of the logPrevalence too.

7. PLOS authors have the option to publish the peer review history of their article (what does this mean?). If published, this will include your full peer review and any attached files.

Reviewer #3: No

---

## [Author Response · Author response to Decision Letter 2]

13 Mar 2024

As you rightly said, there was an error when we wrote the syntax for selogprevalence that caused problems with the result. So considering your comment, we conducted a re-analysis using Stata 18. We used a point-and-click option in Stata 18 rather than writing syntax. We also use proportion without log transformation rather than percentage. There was a significant change in the results on the funnel plot, and some results that we showed with track changes were in a separate document. We would like to thank you again for your valuable and pertinent comments.

---

## [Decision Letter · Decision Letter 3]

17 Mar 2024

PONE-D-23-16216R3Magnitude of Standard Precautions Practices among Healthcare Workers in Health Facilities of Low and Middle Income Countries: A Systematic Review and Meta-analysis.

PLOS ONE

Dear Mengistu Yilma Dilnessie,

Thank you for submitting your manuscript to PLOS ONE. After careful consideration, we feel that it has merit but does not fully meet PLOS ONE’s publication criteria as it currently stands. Therefore, we invite you to submit a revised version of the manuscript that addresses the points raised during the review process:

Please address the comments by the third reviewer It's important to clarify a few points regarding your search to ensure the rigour of your systematic review process:Google Scholar as a Search Engine: Google Scholar is a valuable search engine, but it is not a traditional academic database (as you've claimed). It lacks the controlled indexing, sophisticated search features, and quality filters necessary for systematic reviews. Indeed, your PRISMA flowchart currently reflects only a PubMed search. If you used Google Scholar as a database, then the flowchart needs to be updated to accurately depict your search strategy.Addressing Limitations: Please expand your limitations section to consider the following:What relevant studies might have been missed by not including traditional databases?Could the reliance on PubMed and Google Scholar introduce bias into your study selection?How might this impact the generalizability of your findings?Please submit your revised manuscript by May 01 2024 11:59PM. If you will need more time than this to complete your revisions, please reply to this message or contact the journal office at plosone@plos.org. Please include the following items when submitting your revised manuscript:A rebuttal letter that responds to each point raised by the academic editor and reviewer(s). You should upload this letter as a separate file labeled 'Response to Reviewers'.A marked-up copy of your manuscript that highlights changes made to the original version. You should upload this as a separate file labeled 'Revised Manuscript with Track Changes'.An unmarked version of your revised paper without tracked changes. You should upload this as a separate file labeled 'Manuscript'.If applicable, we recommend that you deposit your laboratory protocols in protocols.io to enhance the reproducibility of your results. Protocols.io assigns your protocol its own identifier (DOI) so that it can be cited independently in the future. For instructions see: https://journals.plos.org/plosone/s/submission-guidelines#loc-laboratory-protocols. Additionally, PLOS ONE offers an option for publishing peer-reviewed Lab Protocol articles, which describe protocols hosted on protocols.io. Read more information on sharing protocols at https://plos.org/protocols?utm_medium=editorial-email&utm_source=authorletters&utm_campaign=protocols.

We look forward to receiving your revised manuscript.

Kind regards,

Isaac Amankwaa, Ph.D.

Guest Editor

PLOS ONE

Journal Requirements:

Reviewers' comments:

Reviewer's Responses to Questions

**Comments to the Author**

1. If the authors have adequately addressed your comments raised in a previous round of review and you feel that this manuscript is now acceptable for publication, you may indicate that here to bypass the “Comments to the Author” section, enter your conflict of interest statement in the “Confidential to Editor” section, and submit your "Accept" recommendation.

Reviewer #3: (No Response)

2. Is the manuscript technically sound, and do the data support the conclusions?

Reviewer #3: (No Response)

3. Has the statistical analysis been performed appropriately and rigorously? 

Reviewer #3: (No Response)

4. Have the authors made all data underlying the findings in their manuscript fully available?

Reviewer #3: (No Response)

5. Is the manuscript presented in an intelligible fashion and written in standard English?

Reviewer #3: (No Response)

6. Review Comments to the Author

Reviewer #3: The funnel plot can be either plotted as log-odds vs 1/standard error or log-odds vs sample size. Please check the paper by Hunter et al for details.

Hunter et al "In meta-analyses of proportion studies, funnel plots were found to be an inaccurate method of assessing publication bias" 2014 Journal of Clinical Epidemiology

7. PLOS authors have the option to publish the peer review history of their article (what does this mean?). If published, this will include your full peer review and any attached files.

Reviewer #3: No

---

## [Author Response · Author response to Decision Letter 3]

20 Mar 2024

The context in Hunter et al. was different from our proportion meta-analysis. They used log-odds vs sample size for a situation when the underlying proportion is quite extreme (very low or very high). The key findings in their study also explained that conventional funnel plots appear to be an inaccurate way of assessing publication bias (PB) in meta-analyses of proportion studies with extreme proportional outcomes. We have tried their suggestion, but the result was unreliable and distorted for our study. 

Moreover, we reviewed further literatures to decide assessment of publication bias in our case. There are arguments about assessing publication bias in proportion meta-analysis. As publication bias was assumed in the context of comparative studies that studies with positive results are more frequently published than studies with negative results, this doesn’t make sense for proportion studies with no comparison. As a result, studies came up with various suggestions that Barker et al. do not suggest using funnel plots, Egger’s and Begg’s tests for proportional meta-analysis, Wang N. suggests to use these tests but interpreting with caution, and Hunter et al. suggest using study size against log odds to assess publication bias in proportion meta-analysis for proportions with extreme outcomes. After careful consideration of this evidence, we decided to use the funnel plot and Egger’s test at least to visualize the distribution of studies and statistical testing along the pooled result. 

We clarified this in the revised manuscript.

We clarified this in the revised manuscript.

As we explained above Hunter et al. key finding explained “conventional funnel plots appear to be an inaccurate way of assessing publication bias (PB) in meta-analyses of proportion studies with extreme proportional outcomes. Thus our proportion was not extreme.

---

## [Editor Report · Decision Letter 4]

2 Apr 2024

Magnitude of Standard Precautions Practices among Healthcare Workers in Health Facilities of Low and Middle Income Countries: A Systematic Review and Meta-analysis.

PONE-D-23-16216R4

Dear Mengistu Yilma,

We’re pleased to inform you that your manuscript has been judged scientifically suitable for publication and will be formally accepted for publication once it meets all outstanding technical requirements.

Kind regards,

Isaac Amankwaa, Ph.D.

Guest Editor

PLOS ONE